# Enhancing Photosynthetic Carbon Transport in Rice Plant Optimizes Rhizosphere Bacterial Community in Saline Soil

**DOI:** 10.3390/ijms252212184

**Published:** 2024-11-13

**Authors:** Weiwei Zhang, Shunying Yang, Tianqi Wei, Yanhua Su

**Affiliations:** 1Institute of Soil Science, Chinese Academy of Sciences, Nanjing 210008, China; 2University of Chinese Academy of Sciences, Beijing 100049, China

**Keywords:** carbon deposition engineering, saline soil, rhizosphere, bacterial structure, complex network

## Abstract

Saline soils exert persistent salt stress on plants that inhibits their ability to carry out photosynthesis and leads to photosynthetic carbon (C) scarcity in plant roots and the rhizosphere. However, it remains unclear how a rhizosphere environment is shaped by photosynthetic C partitioning under saline conditions. Given that sucrose is the primary form of photosynthetic C transport, we, respectively, created sucrose transport distorted (STD) and enhanced (STE) rice lines through targeted mutation and overexpression of the sucrose transporter gene *OsSUT5*. This approach allowed us to investigate different scenarios of photosynthate partitioning to the rhizosphere. Compared to the non-saline soil, we found a significant decrease in soil dissolved organic carbon (DOC) in the rhizosphere, associated with a reduction in bacterial diversity when rice plants were grown under moderate saline conditions. These phenomena were sharpened with STD plants but were largely alleviated in the rhizosphere of STE plants, in which the rhizosphere DOC, and the diversity and abundances of dominant bacterial phyla were measured at comparable levels to the wildtype plants under non-saline conditions. The complexity of bacteria showed a greater level in the rhizosphere of STE plants grown under saline conditions. Several salt-tolerant genera, such as *Halobacteroidaceae* and *Zixibacteria*, were found to colonize the rhizosphere of STE plants that could contribute to improved rice growth under persistent saline stresses, due to an increase in C deposition.

## 1. Introduction

Modern crop production practices are needed to support the world’s growing population, which depends not only on increasing crop yields per unit area but also on the utilization of potentially available land resources, such as saline soils, for extended crop production [1,2]. However, saline soils impose salt stress on crops and lead to severe growth inhibition and yield loss, which also severely inhibits the colonization of some taxa of soil microorganisms. The accumulation of Na^+^ results in a decrease in chlorophyll content in plant leaves, thereby impairing photosynthesis. Consequently, a significant reduction in photosynthetic carbon (C) production is a prevalent issue in plants subjected to stress conditions [3,4,5]. As a result, photosynthetic production scarcity results in a reduction in C deposition.

Sucrose is the major transport form of photosynthetic C, which contributes to the allocation of C to plant grains and roots [6]. In the case of environmental fluctuations that often take place in the field, plants activate energy-consuming protection reactions at the cost of growth and carbon loss. Previous studies have shown that sucrose supply protects plant roots from ammonium toxicity by C replenishment [7,8]. In plants, sucrose is transported and allocated by sucrose transporters named SUC or SUT [9,10,11]. In the rice genome, five SUTs have been annotated and some of them have been studied [9]. *OsSUT5*, a member of rice SUTs, has been shown to be important in mediating long-distance sucrose transport from shoots to roots [10,12,13].

Plant roots connect plants to the soil by absorbing water and nutrients and returning available C and N via exudates and root debris, which determines the microbial activities of the rhizosphere [14,15,16]. However, in saline soils, root growth is straight-forwardly inhibited, causing reduced capability of water and nutrients supply to the above-ground organs that hinders plant growth [17]. In addition, a lowered efficiency of leaf photosynthesis would cause ‘deficiencies’ of plant C as well as its deposition to the rhizosphere. A report has shown that by providing a readily available C source, such as sucrose, plant roots enrich and recruit microbes to the rhizosphere [18]. The capacity of rhizosphere microbial enrichment is related to the capacity of C sources supplied by plant roots [19,20]. To this end, efficient sucrose transport from plant shoots to roots plays a vital role in supporting the bioactivity of the rhizosphere ecosystem. Furthermore, the niche breadth of microbial communities can be classified into generalists and specialists [21,22]. Generalists are normally more resistant to changing conditions, while specialists show greater adaptability to specific niches [23,24,25]. Salinity is an environmental factor of interest that influences the distribution of generalists and specialists [26]. These observations allow us to hypothesize that the enhancement of photosynthetic C deposition to the rhizosphere may facilitate the colonization of certain function microorganisms in the rhizosphere that increase plants’ tolerance to abiotic stresses such as saline conditions.

In this study, we used genetically engineered rice plants with intrinsic differences in sucrose transport activities: distortion (STD) and enhancement (STE) through targeted mutation and overexpression of the shoot–root sucrose transporter gene *OsSUT5* to test this hypothesis in a moderate saline soil.

## 2. Results

### 2.1. Rice Bacterial Community Responses to DOC Under Saline Conditions

DOC showed no change in the bulk of WT (rice without mutation or overexpression) but significantly decreased in the rhizosphere subjected to saline environments (Figure 1a). Bacterial communities’ changes demonstrated that the DOC difference correlated to the rhizosphere bacterial community. When grown in saline conditions, the relative total abundance of dominant phyla (*Firmicutes*, *Chloroflexi*, *Actinobacteriota*, and *Proteobacteria*) and diversity of rhizosphere bacteria were reduced (Figure 1b,c). Since carbon deposition contributed to the DOC in the rhizosphere [27], different sucrose transport scenarios were constructed.

### 2.2. Rhizosphere Bacterial Community Response to Different C Deposition Conditions Under Saline Soil

Figure 2 shows the impacts of sucrose transport and deposition senecios on the rhizosphere microorganisms. Under normal conditions, the photosynthesis parameters and stomatal conductance among all rice plants showed no significant differences. However, after being treated with NaCl, the net photosynthesis parameters in the STD decreased by 52% relative to the WT, while it increased by 33% in the STE (Figure 2a). Stomatal conductance in STD decreased by 21%, while in STE, increased by 8% (Figure 2b). Concurrently, sucrose accumulation was more pronounced in the leaves of STD under salt stress (Figure 2c). Additionally, sucrose transport to the roots in STD decreased by 42% compared to WT, whereas the sucrose content in STE roots increased by 35% (Figure 2d). This trend paralleled the observed changes in dissolved organic carbon (DOC) within the rhizosphere (Figure 2e). Additionally, the soil physicochemical properties and soil enzyme activities of the STD and STE soils under control or saline conditions are listed in Appendix A. Soil inorganic N (NH_4_^+^−N), AP, and AK were reduced in STE under saline conditions (Appendix A). Meanwhile, the soil enzymes such as sucrase, urease, acid phosphatase, and catalase of STE were higher than STD under NaCl treatments (Appendix A). However, these indices showed no significant change in the bulk soil.

After removing chimeras and low-quality sequences, we found that the C levels from shoots significantly influenced the soil microbial community. Our analysis of 4,718,520 bacterial *16S rRNA* sequences identified 65,536 bacterial ASVs (amplicon sequence variants). The dominant phyla across all rice plants and treatments were *Firmicutes*, *Chloroflexi*, *Actinobacteriota*, and *Proteobacteria*. Notably, the relative total abundances of these four phyla were increased in the rhizosphere of STE compared to STD under saline conditions (Figure 2f). The Shannon index of STE rhizosphere was also higher than STD under saline conditions (Figure 2g).

Furthermore, a principal coordinate analysis (PCoA) revealed no distinction in C deposition levels under control conditions (Figure 2h), but a clear distinction under NaCl treatment (Figure 2i). In contrast, no clear distinction was observed in bulk soil (Appendix A).

### 2.3. Bacterial Enrichment Responded to C Deposition

We found that bacterial enrichment responded to the different C deposition conditions positively. Figure 3a showed that 37 depleted and 62 enriched ASVs were observed under control (Figure 3a). However, there were 279 ASVs (115 ASVs depleted and 164 ASVs enriched) under saline conditions (Figure 3b, Log2FC > 2 and *p*< 0.05).

Differential bacteria between STD and STE were further categorized into functional (functioning in C cycle or nutrient availability) and pathogenic (microbes that cause phytopathogenic infections were classified as pathogenic bacteria) groups. Notably, an increase of 83 species of C degradation-related microbes was observed under saline conditions (Appendix A); a list of classifications and references are provided in Appendix A. When treated with NaCl, fewer functional microbes were found in the rhizosphere of STD, but a higher amount in STE (Appendix A, Appendix A). Also, ASVs in DAA at the genus level were screened and classified for generalists/specialists (Appendix A); we observed more generalists under normal conditions (Appendix A), while NaCl treatments promoted the colonization of the specialists (Appendix A).

### 2.4. Bacterial Composition Under Various Sucrose Downward Conditions

To elucidate the species composition of differential bacteria, we identified the top 10 abundant ASVs from the rhizosphere DAA results. The dot plot revealed no significant difference in bacterial abundance between STD and STE under control (Figure 4a). However, under saline conditions, the relative abundances of the enriched bacteria decreased in STD, while STE exhibited an increase in *Micromonosporaceae*, *Peptococcus*, *Lutispora*, *Pedosphaearaceae*, and *Sphingopyxis*, which are involved in the carbon cycle. Notably, salt-tolerant microorganisms, including *Halobacteroidaceae* and *Zixibacteria*, were enriched in STE under saline conditions (Figure 4b).

### 2.5. Enhanced C Deposition Positively Contributed to Bacterial Co-Occurrence Patterns and Rice Growth

The co-occurrence network analysis revealed distinct bacterial interactions [28]. Reduced nodes were found in the rhizosphere (344–521) (Figure 5), in contrast to 529–850 nodes in the bulk soil (Appendix A). Meanwhile, there were fewer connections (edges) in the rhizosphere network (Figure 5). Distorted sucrose transport plants showed fewer nodes and connections and were further reduced under NaCl treatment (Figure 5a,c). Although saline treatment also reduced the network patterns of STE, it remained at a higher level compared to STD (Figure 5b,d). In addition, the rhizosphere network of STE had a stronger correlation compared to STD (Appendix A).

In addition, under normal conditions, the bacterial network of STD and STE did not exhibit significant differences. However, STE, which possesses a more complex bacterial network, demonstrated higher biomass compared to WT and STD (Appendix A).

### 2.6. Correlations Between Soil Microbes and Environmental Factors

The correlation analysis showed that C cycle bacteria, such as *Micromonosporaceae*, *Peptococcus*, and *Lutispora*, were positively related to DOC and soil nutrient assimilation enzymes such as sucrase, functioned in C decomposition (Figure 6a). Interestingly, bacteria that respectively showed tolerance to saline environments, *Patescibacteria* and *Verrucomicrobiota*, were positively related to DOC and Sucrase (Figure 6a).

To better understand the relative contributions of the STE effect to soil properties and *bacterial* enrichment under saline treatment, we constructed the partial least squares path model (PLS-PM) (Figure 6b). Soil DOC showed strong influences on both bacterial community (path coefficient = 0.944) and soil enzyme activity (path coefficient = 0.956). Additionally, the soil DOC level positively influenced the shoot biomass (path coefficient = 0.805). However, the bacterial community (path coefficient = −0.168) exhibited a negative effect on the soil available nutrient change.

## 3. Discussion

Sucrose is a major transport form of plant photosynthetic products, providing the C skeleton and energy for plants, participating in physiological processes and depositing in the rhizosphere [29,30]. Photosynthetically derived C can augment the rhizosphere C pool, thereby facilitating microbial colonization [31]. An enhanced rhizosphere microbial environment can efficiently convert soil nutrients, thereby promoting plant growth. These findings highlight the potential of targeting sucrose transporters to enhance plant tolerance to saline conditions.

In this study, we found that DOC was highly related to the bacterial community under saline conditions (Figure 1). Therefore, we investigated the effects of downward photosynthetic C on the bacterial community under saline conditions using an important rice sucrose transporter (OsSUT5), which functions in sucrose transport from shoots to roots [32]. The results indicate that different sucrose transport scenarios significantly influence the abundance and diversity of the rhizosphere bacterial community under saline conditions, potentially due to C deposition from roots [33,34]. Also, the nutrient availability was promoted in the STE rhizosphere. We demonstrate that the enhancing sucrose transport improved the bacterial colonization in the rhizosphere under NaCl treatments (Figure 2, Figure 6c and Appendix A). Bacteria that function in the C cycle or the N cycle were observed to decrease in STD but were enriched in STE, suggesting that more C delivery to rhizosphere fosters the growth of functional microbes (Figure 4 and Appendix A). Additionally, specialists, characterized by their high reproduction rates and efficient utilization of recalcitrant C sources [21,22], increased in proportion under salt stress and further proliferated in STE. This indicates that the availability of rhizosphere dissolved organic carbon (DOC) plays a pivotal role in shaping the microbial community (Figure 4 and Appendix A), since the low DOC level in the rhizosphere of STD plants limits bacterial proliferation. It is noteworthy that generalists maintained a stable community under control conditions, consistent with previous findings [23,24]. No significant differences were observed in the bacterial community of bulk soil samples; hence, functional bacteria and generalists/specialists were not analyzed or classified for these samples.

Microorganisms within soil ecosystems engage in interactions that collectively mediate biogeochemical transformations and energy transfer [35,36,37]. These microbial interactions can be elucidated through co-occurrence networks [38,39]. In this study, the rhizospheric environment under saline conditions was found to favor the enrichment of functional microbes and enhance the complexity of microbial correlations (Figure 6). The topological parameters of bacterial networks were reduced in the rhizosphere compared to bulk soil, indicating higher levels of interaction, consistent with previous findings [40]. The number of edges and nodes in STD and STE, respectively, decreased and increased, attributable to the distortion and enhancement of endogenous sucrose transport, significantly impacting DOC content in the rhizosphere. These differences were further accentuated under NaCl treatment, suggesting that photosynthetic C supply significantly influences bacterial interrelationships in the rhizosphere. It has been demonstrated that more complex microbial communities exhibit greater stability in response to environmental stresses [41,42]. Our findings indicate that bacteria in the rhizosphere of STE exhibit closer microbe–microbe interactions and promote more efficient nutrient transformation compared to STD.

This study, by conducting distorted/enhanced sucrose transport conditions in rice, extends the significance and impact of photosynthate on the rhizosphere ecosystem. This linkage between the plant C source and the priming of rhizosphere microbes [16,43,44,45], enhances saline tolerance in both aboveground and belowground systems.

## 4. Materials and Methods

### 4.1. Plant Materials

Mutants of *OsSUT5* (STD) and the overexpression of OsSUT5 (STE) generated from a Nipponbare (Nip) background were constructed by our laboratory. For STD, a homozygous mutagenesis of a “T” base addition to the 3rd exon of the *OsSUT5* gene, corresponding to the position 319 of the open reading frame, was used. Sequencing verification of the STD plants are shown in Appendix A, and the expression level of *OsSUT5* in the STE plants increased by 15-fold (Appendix A). Seeds from both the homozygous STD and STE plants were sterilized, germinated, and grown in IRRI nutrient solution (1.25 mM NH_4_NO_3_, 0.3 mM KH_2_PO_4_, 0.35 mM K_2_SO_4_, 1 mM CaCl_2_·2 H_2_O, 1 mM MgSO_4_·7 H_2_O, 0.5 mM Na_2_SiO_3_, 20 μM NaFeEDTA, 20 μM H_3_BO_3_, 9 μM MnCl_2_·4 H_2_O, 0.32 μM CuSO_4_·5 H_2_O, 0.77 μM ZnSO_4_·7 H_2_O, and 0.39 μM Na_2_MoO_4_·2 H_2_O, pH 5.8) for 20 days in a plant growth chamber before being used for pot experiments.

### 4.2. Pot Experiments

Plant growth tests were carried out in pots filled with 3 kg of paddy soil and then mixed with 5.4 g NaCl (added in the treatment group to simulate a saline soil environment with 50 mM final molarity of NaCl) and fertilizers (0.6 g N, 0.45 g P, and 0.45 g K). The pH of the paddy soil used in this study was about 5.6, and soil moisture was kept at about 58%; the other properties are summarized in Appendix A. Four uniform 20-day-old seedlings were transplanted into three replicate pots and grown to full-stage growth in a green house at 26 ± 0.5 °C with a photoperiod of 16 h light and 8 h dark. The soils were irrigated with equal amounts of water as needed. Rhizosphere soils were collected using rhizo-bags (the rhizo-bags were made of nylon mesh with a 30 μm aperture, 7 cm in diameter × 12 cm in height) based on a previously study with slight modifications [46]. Briefly, after the rice roots completely filled the rhizo-bags, the soil within the rhizo-bags was designated as rhizosphere soil, while the soil within a 2 cm radius outside the rhizo−bags was classified as bulk soil. Each rhizo-bag contained a single seedling.

### 4.3. Plant and Soil Analysis

The photosynthetic parameters of flag leaves were measured at the early tillering stages using a Li-6400 portable photosynthesis analyzer (Li-Cor, Lincoln, NE, USA). Determination was conducted from 9:00 to 11:00 during sunny mornings in the following working conditions: light intensity of 1000 μmol m^−2^ s ^−1^, CO_2_ concentration set at 400 μmol mol^−1^, air flow rate at 500 μmol s^−1^, and temperature set at 28 °C.

The flag leaves and roots of WT, STD, and STE were sampled at the tillering stage and used to determine the sucrose contents with a sucrose measurement kit (Nanjing Jiancheng Bioengineering Institute, Nanjing, China) according to the manufacturer’s instructions.

Rhizosphere and bulk soil samples were collected according to the method of Nie et al. [46] at 35 days post-transplantation, since roots filled the rhizo-bag at this stage. Soil filled in the rhizo-bag was collected as the rhizosphere sample, while soil distal from the rhizo-bag was treated as bulk soil. To assess the soil enzyme activity, the sucrase, urease, catalase, and acid phosphatase of the rhizosphere and bulk soil were assessed respectively by the 3,5−dinitrosalicylic acid colorimetric method, sodium salicylate-sodium dichloroisocyanurate colorimetric method, potassium permanganate titration method, and p-nitrophenyl disodium phosphate colorimetric method [47]. For the urease activity assay, a five-gram sample of air-dried soil was placed in a 250 mL flask, and 2.5 mL of 0.08 M urea solution was added. The flask was sealed and incubated at a constant temperature of 37 °C for 2 h. Then, 2.5 mL of distilled water and 50 mL of 2 M KCl solution were added, shaken at 180 rpm for 30 min at room temperature, and then filtered. Then, 1 mL aliquots of the filtrate was mixed with 9 mL of distilled water. Subsequently, 2.5 mL of the solution (0.3 M NaOH: 1.06 M sodium salicylate = 1:1) and 1 mL of 39.1 mM sodium dichloroisocyanurate solution were added in sequence to a 25 mL test tube. The released NH_4_^+^−N was determined calorimetrically at the 660 nm wavelength, and the unit was expressed as an amount of mg·g^−1^. For the phosphatase activity assay, one gram of air-dried soil was spiked with 1 mL of toluene for 15 min in a 50 mL conical flask with a stopper and then mixed with 20 mL of phosphate buffer and 5 mL of 0.05 M p-nitrophenyl phosphate disodium solution. After incubating at 37 °C for 1 h, 5 mL of 0.5 M calcium chloride solution and 20 mL of 0.5 M sodium hydroxide were added and centrifuged at 2500 rpm for 5 min. The supernatant (5 mL) was further centrifuged at 3800 rpm for 5 min, the absorbance at 410 nm was recorded, and the unit was expressed as the amount of mg·g^−1^. For the sucrase activity assay, a five-gram sample of air-dried soil was mixed with 15 mL of an 8% sucrose solution, 5 mL of phosphate buffer (pH 5.5), and 1 mL of toluene in a 50 mL conical flask and filtered after incubating at 37 °C for 24 h. Then, 1 mL of the filtrate was mixed with 3 mL of 3,5−dinitrosalicylic acid in a new 50 mL conical flask, heated in a water bath for 5 min, and then cooled for 3 min. The absorbance of 508 nm was recorded, and the unit was expressed as an amount of mg·g^−1^. Catalase activity was determined by the potassium permanganate titration volume method and expressed as 0.1 M potassium permanganate in 1 g of soil after 20 min (mg·g^−1^·20 min^−1^).

The contents of dissolved organic carbon (DOC) in air-dried soils were determined by oxidation with the potassium dichromate-colorimetric method. Soil inorganic N (NH_4_^+^−N), nitrate N (NO_3_^−^−N), and available phosphorus (AP) and potassium (AK) were extracted with 2 M KCl, 0.5 M NaHCO_3_ (pH 8.5), and 1 M NH_4_OAc, respectively, determined by an automated chemical analyzer (Smartchem 200, AMS, Barcelona, Spain), molybdenum antimony chronometry method, and a flame spectrophotometer (M410, Sherwood, UK).

### 4.4. Microbial Analysis

DNA was extracted from the rhizosphere and bulk samples using the DNA TM SPIN Kit for Soil (MP Biomedicals, Santa Ana, CA, USA) according to the manufacturer’s protocols. Bacterial sequencing libraries were prepared from 72 DNA samples. The V3-V4 region of the bacteria 16S ribosomal RNA gene was amplified by PCR (95 °C for 5 min, followed by 25 cycles at 95 °C for 30 s, 55 °C for 30 s, and 72 °C for 30 s, and a final extension at 72 °C for 5 min) using primers 515F (5′−GTGCCAGCMGCCGCGG−3′) and 907R (5′−CCGTCAATTCMTTTRAGTTT−3′) [48]. Each PCR reaction (20 µL) contained 10 µL of 2 × Phanta Max Master Mix (Vazyme, Nanjing, China), 1 µL of template DNA (~50 ng), and 0.4 µL of each primer (10 pmol/µL). Amplicons were extracted from 2% agarose gels and purified using the AxyPrep DNA Gel Extraction Kit (Axygen Biosciences, Union City, CA, USA) according to the manufacturer’s instructions. Purified PCR products were quantified using NanoDrop One (ThermoFisher, Waltham, MA, USA). Amplicon libraries were sequenced by a commercial company on the Illumina HiSeq2500 platform (Majorbio Company, Shanghai, China). Copy numbers of soil microbes were analyzed using QuantStudio3 with conserved primers. Primers and procedures of the real−time quantitative PCR (qRT−PCR) analysis are shown in Appendix A.

The 16S rRNA gene sequencing data have been deposited in the National Center for Biotechnology Information (NCBI, Bethesda, MD, USA) Sequence Read Archive under accession number PRJNA1060240. Sequencing data were analyzed using QIIME2 (Quantitative Insights Into Microbial Ecology 2, https://qiime2.org/, accessed on 10 November 2024) software [49]. Primer sequences were trimmed with the cutadapt plugin [50], and the DADA2 plugin was employed for filtering, denoising, and chimera removal [51]. This process yielded 4,718,520 high-quality reads after excluding short, low-quality reads, singletons, triplicates, and chimeras. Taxonomic assignment of amplicon sequence variants (ASVs) was performed using the classify−sklearn naive Bayes classifier [52] with the SILVA 132 database for bacteria.

Alpha diversity indices (e.g., ACE and Shannon index) were calculated using QIIME 2. We performed a permutational multivariate analysis of variance (PERMANOVA) with the adonis function in vegan R package [53] to assess the effects of different factors on the community dissimilarity, using 999 permutations and Bray–Curtis distance matrix as an input. A principal coordinate analysis (PCoA) was performed based on Bray–Curtis dissimilarity matrix using the “vegan” package to visualize the differences in the bacterial communities in soil samples [54].

We conducted a differential abundance analysis (DAA) to explore the most discriminant ASVs among sucrose transport scenarios under control or NaCl treatments. Briefly, we used the glmFit function to fit a generalized linear model (GLM) to the read counts; ASVs were then screened by likelihood ratio tests using the edgeR package using a trimmed mean of M-values (TMM) normalization method and a threshold of significance at *p* < 0.001 [55].

Generalists or specialists were identified when they exceeded the upper limit of the 95% confidence interval or fell below the lower limit of the 95% confidence interval for the 1000 alignments [56].

### 4.5. Co-Occurrence Network and Correlation Analysis

We used the R packages “Hmisc” and “tidyfst” to calculate the co-occurrence network of bacterial communities, with relative abundance (RA) > 0.5%, significant correlation *p* < 0.05, and Spearman coefficient N > 0.6 or <−0.6 [57]. Also, a network topology domain analysis was conducted based on the average degree. The results of the co-occurrence network were visualized with R packages “ggplot2” and “Ggnewscale”. Correlations between rhizosphere bacterial communities and soil physicochemical properties were assessed with R packages “corrplot” and “pheatmap”.

### 4.6. Data Analysis

The data were processed using Microsoft Excel, Sigmaplot, and R software. Significances were analyzed with a two−way analysis of variance (ANOVA). PCoA, volcano plot, co-occurrence network, and correlation analyses were carried out using the R software v3.6.3. PLS−SEM analyses were conducted using the Smart PLS 3.0 software (SmartPLS GmbH, Boenningstedt, Germany) to reveal the effects on phloem sap sucrose content, soil DOC content, bacterial community (bacterial diversity), soil available nutrient (NH_4_^+^−N, AP, and NO_3_^−^−N), and soil enzyme activity (sucrase, catalase, acid phosphatase, and urease).

## 5. Conclusions

The present work, by molecular manipulation of a single sucrose transporter, OsSUT5, extends the importance of photosynthetic carbon transport in plant growth to its influences on the rhizosphere ecosystem, linking the plant carbon source to the priming of rhizosphere microbes, creating a favorable rhizosphere microenvironment, and thus promoting the rice growth under saline conditions.

## Figures and Tables

**Figure 1 ijms-25-12184-f001:**
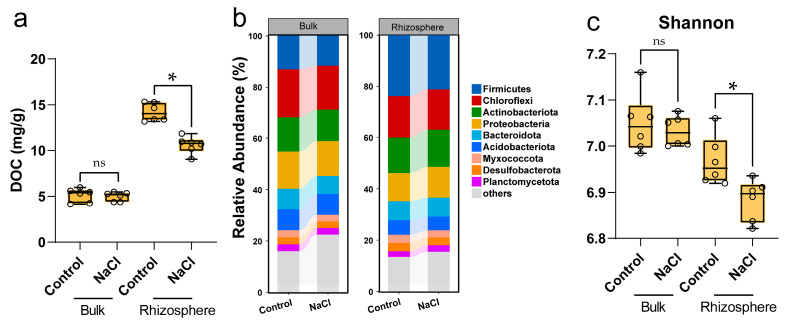
Different soil DOC levels affect bacterial abundance and α-diversity. (**a**) Soil (DOC) dissolved organic carbon level of WT plants under control or saline conditions; (**b**) telative abundance of bacterial phyla (**a**) and alpha diversity analyses, as well as Shannon (**c**) in rhizosphere and bulk bacteria of WT rice plants. One-way ANOVA; *: *p* ≤ 0.05; ns: *p* > 0.05.

**Figure 2 ijms-25-12184-f002:**
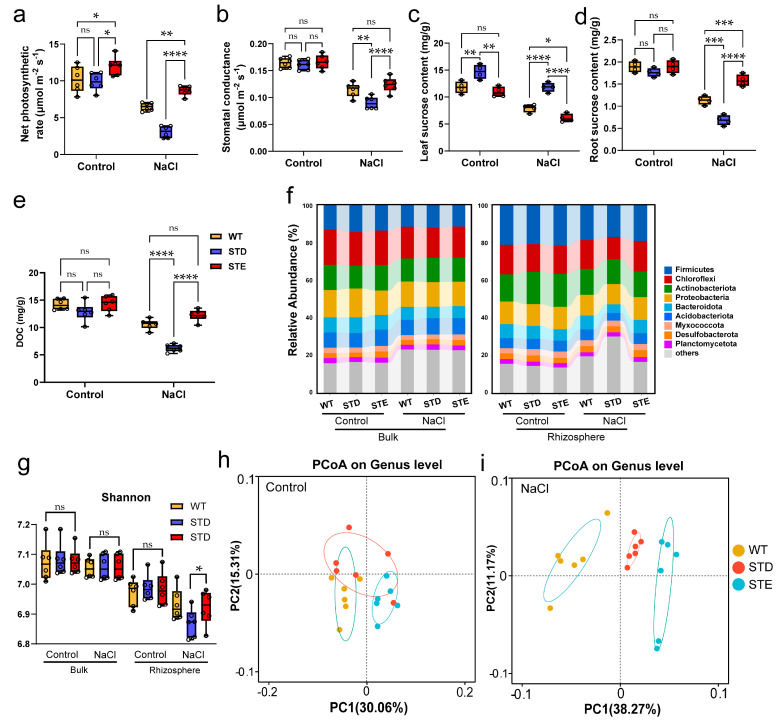
Rhizosphere bacterial community response to different carbon allocation and deposition conditions. Net photosynthetic rate (**a**), stomatal conductance (**b**), leaf sucrose content (**c**), root sucrose content (**d**), and soil dissolved organic carbon (DOC, (**e**)) of WT, STD, and STE under control or NaCl treatment. Relative abundance of bacterial phyla (**f**) and alpha diversity analysis as well as Shannon (**g**) in rhizosphere and bulk bacteria of STD or STE rice plants. The principal coordinate analysis (PCoA) shows microbial community dissimilarity (Bray−Curtis distance) among rhizosphere samples from the three different sucrose transporting circumstances under control ((**h**), R^2^ = 0.6329, *p* = 0.001) or salt stress ((**i**), R^2^ = 0.6883, *p* = 0.001). One-way ANOVA; *: *p* ≤ 0.05; **: *p* ≤ 0.01; ***: *p* ≤ 0.001; ****: *p* ≤ 0.0001; ns: *p* > 0.05.

**Figure 3 ijms-25-12184-f003:**
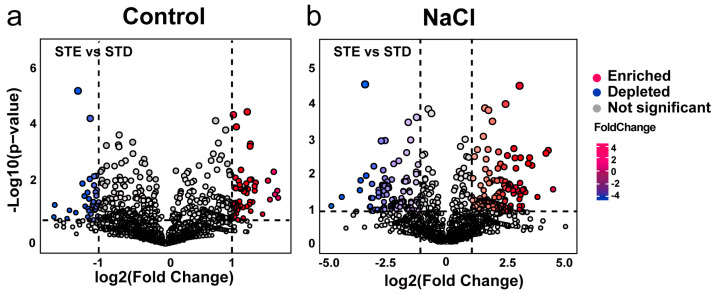
Carbon deposition conditions affect bacterial enrichments. Volcano plots show enrichment and depletion patterns in rhizosphere microbes between STD and STE under control (**a**) or NaCl treatments (**b**).

**Figure 4 ijms-25-12184-f004:**
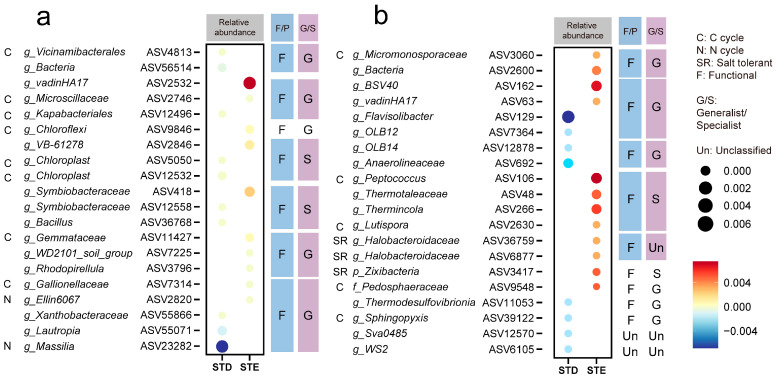
Community change in differential abundance analysis (DAA). Dot plot showing the abundance of enriched and depleted rhizosphere ASVs of control (**a**) and NaCl (**b**) treatment across native, STD, and STE groups. Color intensity corresponds to the relative abundance of specific ASVs. F/P: functional/pathogen microbes, G/S: generalist/specialist, Un: unclassified.

**Figure 5 ijms-25-12184-f005:**
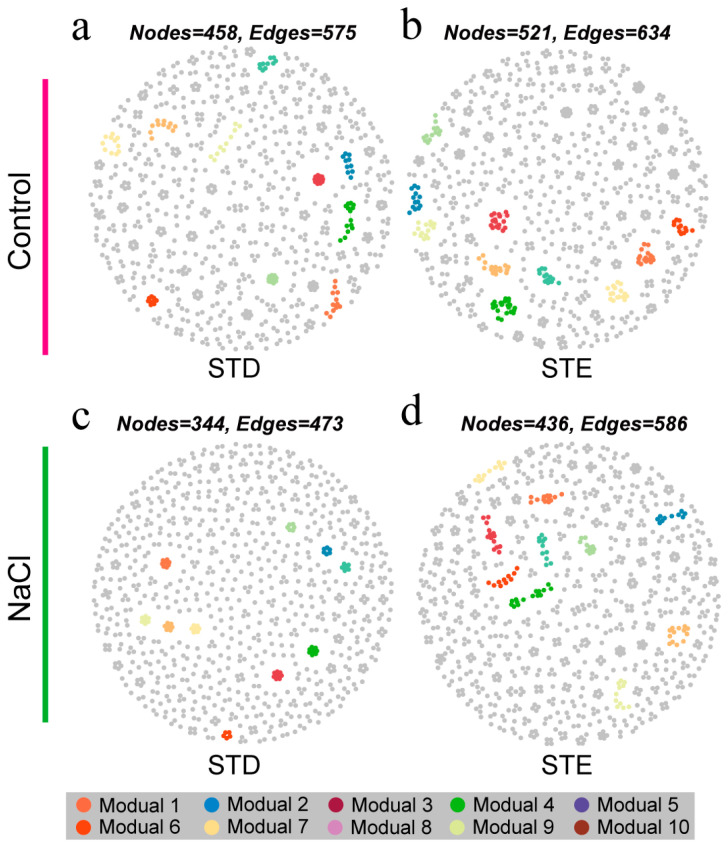
Enhanced carbon deposition positively contributes to bacterial network interactions. Visualization of constructed networks in STD and STE under control (**a**,**b**) and NaCl treatment (**c**,**d**). Different modules are shown in different colors.

**Figure 6 ijms-25-12184-f006:**
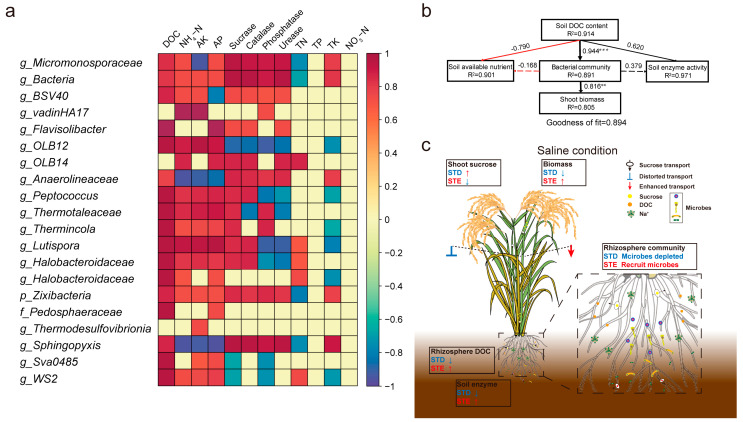
Differential bacteria correlate to environmental correlations and the sucrose transport engineering model. (**a**) Top 10 most abundant taxa in the DAA analyses and their environmental preferences (e.g., positive or negative). (**b**) PLS-PM of the drivers of sucrose content. Defense enzyme activities include CAT, SOD, and POD activities. Soil available nutrient included soil NH_4_^+^−N, AK, and AP. Each oblong box represents a latent variable, which was chosen according to the correlations among these indicators. Path coefficients were calculated after 1000 bootstraps. The black and red lines represent positive and negative effects, respectively. The full and dashed lines indicate the significant correlations (*p* < 0.05) and no correlations (*p* > 0.05), respectively. (**c**) Model of sucrose transporting affects the rhizosphere micro-ecosystem and bacterial community. DOC: dissolved organic carbon; NH_4_^+^−N and NO_3_^−^−N: available nitrogen content; AK: available potassium content; AP: available phosphorus content. Significance level: **: *p* ≤ 0.01; ***: *p* ≤ 0.001.

## Data Availability

Data is contained within the article or Appendix A.

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
