# Peer review of "Enhancing Photosynthetic Carbon Transport in Rice Plant Optimizes Rhizosphere Bacterial Community in Saline Soil"

_ijms, 2024, doi:10.3390/ijms252212184_

Round 1

Reviewer 1 Report

Comments and Suggestions for Authors

In saline soils, plants are subjected to prolonged salt stress, which negatively affects their growth. In this study, the authors generated rice lines with altered sucrose transport activity by targeting-mutation and overexpressing the sucrose transport gene OsSUT5, and used them to investigate the effect of the distribution of photosynthetic products to the rhizosphere on the microbial community. As a result, the STD line, which had a mutation in OsSUT5, showed a decrease in DOC and bacterial relative abundance in saline soil environments. On the other hand, the STE line overexpressing OsSUT5 showed rhizosphere bacterial diversity and abundance of dominant species similar to the non-saline environment.

Salt tolerance is an extremely important issue in agriculture, and the interaction between plants and their microbiomes is also a topic of great interest. I believe that this study, which investigated the effect of transport of photosynthetic products to the rhizosphere on the rhizosphere microbiome, will be of great interest. However, there are several issues that need to be addressed as listed below.

1. The phenotypic analysis of the OsSUT5 knockout and overexpression lines is inadequate.

The premise of this study is that sucrose transport to the rhizosphere is altered in the OsSUT5 knockout and overexpression lines. However, data on photosynthetic activity, sucrose accumulation, etc. are completely lacking for the WT, OsSUT5-KO, and OsSUT5-OE lines. Data on sucrose content in the roots of these three lines are essential, and without these data, the results of this study cannot be discussed.

2. Regarding the changes in DOC

The main analysis in this study is to relate changes in DOC to changes in the microbial community. However, the data for DOC in STD and STE are only shown in the Supplementary Data. They should be included in Figure 2. Also, only STD and STE are shown in Figure 2, but the data for WT are needed. Furthermore, if the data for WT in Figure 1 and STE in Table S3 are comparable, then the conclusion of this paper (that OsSUT5-OE increases DOC and promotes bacterial community) cannot be reached because STE does not increase DOC compared to WT.

3. I don't understand the meaning of the sentence in L167-169.

4. In Figure S1, two lines of OsSUT5-KO and OsSUT5-OE were produced, but which lines were used for STD and STE? If only one line was used, there is no need to show two lines in Figure S1. If both lines were used, the results for each line should be shown.

Reviewer 2 Report

Comments and Suggestions for Authors

The paper has a strong focus and addresses how C flux influences soil microbiomes.   

The work is solid  but this reviewer  requests  more details in methods and legends. This request also is to make the supplemental materials more useful

please see comments as sticky notes   for potential changes

Comments on the Quality of English Language

minor changes

Reviewer 3 Report

Comments and Suggestions for Authors

The Author's used mutants of rice that could shunt more carbon to the roots as compared to the wild type to determine if it would help plant growth and the rhizosphere bacterial community in normal and high salinity conditions. The authors have completed a lot of work but there are several issues that need to be addressed.

1) The main experiment is somewhat straightforward but there is some confusion while reading the experimental design and overall results comparison.  The authors appear to have three kinds of plants: 1) Wild type with no mutation 2) STD (mutant), and 3) STE (Overexpression). I am struggling to see figures and analyses that look at all three except for Fig. S7. As for the STD mutant, there is no information on the expression levels in Fig S1 b so did the mutation show no expression? The figures and analyses should show all three of these together.

Figure 1 shows the differences of the wild type to salt and no salt and soil location, while Figure 2 shows the differences between the STD and STE for salt and soil location. It is hard to tell if these mutates are indeed preforming better when they are not next to each other on the same graph and in one analysis.

2) This manuscript has a separate results and discussion section, so all interpretation needs to be removed from the results and sentences should not be vague. An example of vague are lines (167-168) as notable changes can occur but are they significant. If not, they cannot be notable, but inconclusive. Examples of interpretation include lines (184-185; 193-195, 210-2011; 216-217; 245-247). All of these are suggested reasons and not results. Please move to the results section.

3) Methods and materials need additional information particularly where there are modifications. The authors must note the minor modifications in this manuscript. Also, the methods need to be stand-a-lone. This means that there should be enough information to complete all major steps without going to the citation. Examples of modification are lines (66-67; 80). Examples of insufficient methods are lines (Verification steps (line 67); sample collection (line 85); greenhouse conditions (line 78).

Minor line issues:

Keywords: All title words are keywords so there is no reason to repeat them in the keywords section, please replace photosynthetic carbon with another keyword.

Line 28: This first sentence needs to be altered. Suggest something like" Modern crop production practices are needed to support the worlds growing population...."

Line 42-44: Sentence is awkward as written, please revise.

Line 45: Suggest: Plant roots connect plants to the soil by absorbing water and nutrients and returning available C via exudates and root debris, 

Line 56: suggest: ....may contribute to the colonization of important microorganisms in the rhizosphere of rice."

Line 65, 166, 241: Do not start a sentence with an abbreviation or acronym, please spell it out.

Line 66: Suggest: "...with minor revisions."

Line 94: Please add the make and model of the automated chemical analyzer.

Line 108-110: Did the company or the authors construct the libraries? If the authors completed the library, then they should discuss the barcoding step.

Line 125: It is unlikely the authors completed 1999 permutations, it's more likely it was the standard 999 permutations.

Line 152: Suggested lower case for Bacterial community...

Figure 2: I assume the R2 and P values within c and d are adonis results so it will need to be added to the figure legend. 

Line 190: "for 52%-66% phyla" - This is a little confusing so please correct this for better clarify.

Line 229: Suggest: "...are involved in the carbon cycle."

Line 244-245: Suggest: ... had a stronger correlation"

Lines 274-275: This result is not very straightforward, please rewrite to make it clearer.

Figure S3: R2 and P values are likely adonis analysis so add to the figure legend

Table S3: The table looks at three 3 types of rice plants but there are only two shown. Please complete the table.

Table S6: Suggest: Network patterns in the rhizosphere and bulk soil between rice plants. 

Please take your time with the revisions.

Round 2

Reviewer 1 Report

Comments and Suggestions for Authors

The Revised manuscript addresses many of my previous concerns. However, some concerns remain regarding the new data.

The Net photosynthetic rate in Figure 2a is not explicitly stated under what conditions it was measured. Their results are significantly lower than previously reported photosynthetic rates of Nipponbare (ex. Li et al. Nature Plants, 2020, 5, 848-859, and many other studies measuring photosynthetic rates of Nipponbare). It is not possible to determine whether this is because the measurement conditions are very different, or whether the authors' photosynthesis measurement technique is inexperienced and unreliable. It is essential to indicate the measurement equipment material, light and CO2 concentrations, time of day, and leaf position at which the measurements were taken, and still more, values such as stomatal conductance should also be indicated. Also, a detailed description of the measurement method for sucrose content in Figure 2b, c is required.

Reviewer 3 Report

Comments and Suggestions for Authors

The authors have addressed my concerns, and I think the manuscript is more robust in this version.

There are two minor issues for them to review:

Line 67-38: please check as it likely increases saline-alkali tolerance or reduces saline-alkali stress. As written " At the same time, it can effectively reduce pH value and induce the enrichment of beneficial plant microbial communities, thus reducing the saline-alkali tolerance [11].

Lines 282-283: It appears that this sentence is a repeat of the one above it. Please review.
